# Spatial variations and associated factors of knowledge of ORS packet or pre-packaged liquids for the management of diarrhea among women of reproductive age in Ethiopia: A spatial and multilevel analysis

**Achamyeleh Birhanu Teshale** *, **Getayeneh Antehunegn Tesema, Zemenu Tadesse Tessema**

Department of Epidemiology and Biostatistics, Institute of Public Health, College of Medicine and Health Sciences, University of Gondar, Gondar, Ethiopia

* achambir08@gmail.com

## Abstract

### Background

Even though diarrhea is not lethal by itself, the lack of knowledge about its management results in devastating complications such as dehydration and lastly death. Using an oral rehydration solution (ORS) is an easy, inexpensive, and most reliable way of treating dehydration and reducing diarrhea-related mortalities. The literature revealed that Knowledge of ORS packet or pre-packaged liquids is a very important and critical factor for the utilization of ORS during the management of diarrhea.

### Methods

We used the 2016 Ethiopian Demographic and Health Survey data, which is the fourth survey conducted by the Central Statistical Agency. A total weighted sample of 7590 reproductive-age women who gave birth within five years preceding the survey was used. Multilevel logistic regression analysis was done to assess factors associated with knowledge of ORS packet or pre-packaged liquids. Arc GIS version 10.3 and Kuldorff's SaTScan version 9.6 software were used for the spatial analysis.

### Results

In the multilevel analysis maternal education, media exposure, residence, community illiteracy level, and region were significantly associated with knowledge of ORS packet or pre-packaged liquids. Besides, knowledge of ORS packet or pre-packaged liquids for the management of diarrhea was not random in Ethiopia (with Moran's Index = 0.46 and p-value <0.001), and the primary clusters spatial window was located in SNNPR, most parts of the Oromia region, and eastern parts of the Gambela region.

**Data Availability Statement:** It is ethically not acceptable to share the DHS data sets to third parties. However, anyone who want the data set

can access from the Measure DHS program at www.dhsprogram.com, through legal requesting. The authors had no special access privileges others would not have.

**Funding:** The author(s) received no specific funding for this work.

**Competing interests:** The authors have declared that no competing interests exist.

**Abbreviations:** AOR, Adjusted Odds Ratio; CI, Confidence Interval; EDHS, Ethiopian Demographic and Health Surveys; ICC, Intra-class Correlation Coefficient; MOR, Median Odds Ratio; ORS, Oral Rehydration Solution; PCV, Proportional Change in Variance; VIF, Variance Inflation Factor.

## Conclusion

In this study knowledge of ORS packet or pre-packaged liquids was not random across the country. Lack of formal education, lack of media exposure, being from a rural area, and being from communities with a higher illiteracy level was associated with lower odds of knowledge of ORS packet or pre-packaged liquids. Therefore, special emphasis should be given to these high-risk groups and the hot spot regions (SNNPR, most parts of the Oromia region, and eastern parts of the Gambela region). Moreover, distributing information through different media regarding ORS packet or pre-packaged liquids is necessary.

## Background

Diarrheal disease is the second most common cause of death in under-five children. Each year, there are about 1.7 billion cases of diarrhea and 525,000 deaths due to diarrhea among under-five children worldwide. It is also a leading cause of malnutrition [1]. Diarrheal diseases in under-five children can be intervened at both primary (using proper sanitation and water quality) and secondary prevention (through early recognition of dehydration and prompt administration of oral rehydration solution (ORS)) levels [2].

The use of oral rehydration therapy and zinc, along with continued feeding, for the management of diarrhea is under the Integrated Management of Childhood Illness [3]. Via water replenishment and the replacement of electrolytes in the body, ORS prevents dehydration. It is an easy, inexpensive, and most reliable way of treating dehydration and reducing diarrhea-related complications and mortality [4].

Knowledge of ORS is a very important and critical factor for its utilization for the management of diarrhea in under-five children [5–9]. As revealed by different scholars, it varies across regions ranging from 62% in India to 98% in Nigeria [10, 11]. In Ethiopia, according to the Ethiopian Demographic and Health Survey (EDHS) 2016 report, 66% of reproductive age women knew ORS packets or pre-packaged liquids [4]. Factors such as maternal age, maternal education, media exposure, residence, household wealth status, region, and religion are among the different factors that are associated with knowledge of ORS packet or pre-packaged liquids [11–16].

Even though diarrhea is not lethal by itself, the lack of knowledge about its management can result in devastating complications such as dehydration and lastly death [17, 18]. ORS is one of the most effective treatments for dehydration that happened due to diarrhea and nationwide there were no studies conducted about factors associated with knowledge of ORS packet or pre-packaged liquids. Also, in previous studies, community-level factors that might have a relation with knowledge of ORS packet and pre-packed liquids were not addressed. Moreover, its spatial variation that helps to identify hot spot areas of knowledge of ORS packet or pre-packaged liquids was not assessed before. Therefore, this study aimed to assess spatial distribution and factors associated with knowledge of ORS packet or pre-packaged liquids for the management of diarrhea among women of reproductive age in Ethiopia. The findings of this study could help policymakers and other responsible bodies by identifying the community-level factors, in addition to the individual level factors, and hot spot areas of knowledge of ORS packet or pre-packaged liquids.

## Methods

### Data source

We used the 2016 EDHS data that is the fourth survey conducted by the Central Statistical Agency. It was conducted from January 18, 2016, to June 27, 2016. It is based on a nationally representative sample in which the target groups were women aged 15–49 and men aged 15–59 in randomly selected households across Ethiopia. The Population and Housing Census conducted in 2007 was used as the sampling frame for this survey. Regarding the sampling technique, the EDHS sample was stratified and selected in two stages. Six hundred forty-five Enumeration areas were selected at the first stage and a fixed number of 28 households per cluster were selected at the second stage [4]. For this study, we used a total weighted sample of 7590 women of reproductive age who gave birth within five years preceding the survey.

### Study variables

**Outcome variable.** The outcome variable was Knowledge of ORS packet or pre-packaged liquids. Women had knowledge about ORS packet or pre-packaged liquids if she heard about it or she used it for the management of diarrhea otherwise the women had no knowledge [4].

**Explanatory variables.** We incorporated both individual and community level variables for assessing knowledge of ORS packet or pre-packaged liquids. The individual-level explanatory variables included were maternal age, maternal education, marital status, maternal employment, household wealth index, sex of household head, perception of distance from the health facility, religion, media exposure, and the number of under-five children.

We incorporated five community-level variables. Of these three variables (community women illiteracy level, community level of media exposure, and community poverty level) were aggregated community-level variables. However, the two community-level variables residence and region were non-aggregated community-level variables. Community women's illiteracy level was the proportion of women with no formal education at the cluster level, which was derived from data on respondents' level of education. The community poverty level was the proportion of women in the poorest and poorer quintiles at the cluster level. Community-level media exposure was the proportion of women who had been exposed to at least one media (either television, radio, or newspaper). The aggregated community-level variables were categorized as low and high based on a national median value.

### Data management and statistical analysis

The data was extracted and recoded using Stata version 14 software. To restore the representativeness and get an appropriate estimate (such as a robust standard error), every analysis was based on weighting.

**Multilevel analysis.** Due to the hierarchical nature of the EDHS data, a multilevel logistic regression analysis was done to assess factors associated with knowledge of ORS packet or pre-packaged liquids. Four models were fitted while conducting the multilevel analysis. The first model (null model) was fitted without any explanatory variables. The second model (model 1) was fitted with individual-level variables. The third model (model 2) was fitted using community-level variables and the fourth model (model 3) was fitted using both individual and community level variables simultaneously. In the random effect analysis, the intra-class correlation coefficient (ICC), median odds ratio (MOR), and a proportional change in variance (PCV) were assessed and reported. Finally, model fitness was checked using deviance. In addition, a bivariable analysis was done and those with a p-value less than 0.20 were eligible for the multivariable analysis. Finally, the adjusted odds ratio (AOR) with a 95% Confidence interval was

reported and variables with p value<0.05 were declared to be significant predictors of knowledge of ORS packet or pre-packaged liquids. Moreover, to assess the Multicollinearity between independent variables we used variance inflation factor (VIF) and there was no Multicollinearity (the mean VIF was less than five).

**Spatial analysis.** Arc GIS version 10.3 and Kuldorff's SaTScan version 9.6 software were used for the spatial analyses (spatial autocorrelation, spatial interpolation, hot spot analysis, and SaTScan analysis).

*Spatial autocorrelation*. Global Moran's I statistic was done to ascertain whether the spatial distribution of knowledge of ORS packet or pre-packed liquids was clustered, dispersed, or random across Ethiopia [19].

*Spatial interpolation*. To predict the prevalence of mothers with no knowledge of ORS packet or pre-packaged liquids in the un-sampled areas, we conducted the spatial interpolation using the Kriging spatial interpolation method [20].

*Hot spot and cold spot analysis*. To identify areas with higher rates of mothers with no knowledge about ORS packet or pre-packaged liquids (specific significant hot spots areas) and areas with lower rates of mothers with no knowledge of ORS packet or pre-packaged liquids (cold spot areas), hot spot and cold spot analysis were done using Getis-Ord Gi* statistics [21, 22].

*A spatial scan statistical analysis*. We conducted the Bernoulli based spatial scan statistical analysis to identify significant primary and secondary clusters [23]. While conducting the SaTScan analysis our cases were mothers with no knowledge of ORS packet or pre-packaged liquids and mothers who had knowledge of ORS packet and pre-packaged liquids were considered as our controls. Besides, the coordinate files (latitude and longitude) were required while doing the analysis. We used the maximum spatial cluster size of < 25 percent of the population as the upper limit for the identification of both small and large clusters and missed clusters containing more than the maximum limit. The clusters (both the primary and secondary clusters) were identified and the 999 Monte Carlo replications were used to assign p values and to rank using their log likely hood ratio (LLR) test. The circle or the spatial window with the highest LLR test was the primary cluster (most likely cluster), the cluster that is least likely to occur by chance.

## Ethical consideration

Since we were using the publicly available data, ethical approval was not required as such. But by registering or online requesting we have accessed the data set from the DHS website (https://dhsprogram.com/). Besides, the Institute of Public Health, College of Medicine and Health Sciences, University of Gondar, Institutional Review Committee have deemed this study exempt.

## Results

### Sociodemographic characteristics of respondents

The total sample size was 7590 (weighted) women of reproductive age who gave birth within five years preceding the survey. The median age of the participants was 28 years with an IQR of 24–34 years. The majority of them were in the age group 25 to 34 years. Around two-thirds (63.12%) of the participants had no formal education. The majority, 71.63% of respondents were employed during the time of the survey. About two thirds (65.47%) of the study participants were not exposed to any media. Regarding the place of residence, the majority (87.23%) of the study participants were rural dwellers. Moreover, the majority of the respondents were from Oromia (41.23), Amhara (21.50), and Southern Nation Nationality and People's Region (SNNPR) (21.09%) respectively (Table 1).

**Table 1. Sociodemographic characteristics of study participants/respondents.**

| Variables | Frequency [N = 7590] | Percentage |
|---|---|---|
| Maternal age (years) | | |
| 15–24 | 1804 | 23.77 |
| 25–34 | 3827 | 50.42 |
| 35–49 | 1959 | 25.82 |
| Marital status | | |
| Currently married | 7109 | 93.66 |
| Currently not married | 481 | 6.34 |
| Maternal education | | |
| No formal education | 4791 | 63.12 |
| Primary education | 2150 | 28.32 |
| Secondary education | 420 | 5.53 |
| Tertiary & above education | 229 | 3.02 |
| Religion | | |
| Orthodox Christian | 2882 | 37.97 |
| Protestant | 1652 | 21.76 |
| Muslim | 2824 | 37.21 |
| Others | 232 | 3.06 |
| Household wealth status | | |
| First | 1652 | 21.76 |
| Second | 1654 | 21.79 |
| Middle | 1588 | 20.93 |
| Fourth | 1427 | 18.80 |
| Higher | 1269 | 16.72 |
| Employment status | | |
| Employed | 2172 | 28.62 |
| Not employed | 5418 | 71.38 |
| Distance from the health facility | | |
| Big problem | 4407 | 58.06 |
| Not a big problem | 3183 | 41.94 |
| Number of under five children | | |
| 0 | 282 | 3.71 |
| 1–2 | 6525 | 85.96 |
| 3–6 | 783 | 10.32 |
| Sex of household head | | |
| Male | 6474 | 85.29 |
| Female | 1116 | 14.71 |
| Media exposure | | |
| No | 4969 | 65.47 |
| Yes | 2620 | 34.53 |
| Residence | | |
| Urban | 969 | 12.77 |
| Rural | 6621 | 87.23 |
| Region | | |
| Tigray | 537 | 7.08 |
| Afar | 71 | 0.94 |
| Amhara | 1632 | 21.50 |
| Oromia | 3130 | 41.23 |

(*Continued*)

**Table 1.** (Continued)

| Variables | Frequency [N = 7590] | Percentage |
|---|---|---|
| Somalia | 269 | 3.54 |
| Benishangul | 81 | 1.06 |
| SNNPR | 1601 | 21.09 |
| Gambela | 21 | 0.27 |
| Harari | 17 | 0.23 |
| Addis Ababa | 198 | 2.61 |
| Dire Dawa | 33 | 0.44 |
| Community poverty level | | |
| Low | 4529 | 59.67 |
| High | 3061 | 40.33 |
| Community illiteracy level | | |
| Low | 3846 | 50.67 |
| High | 3744 | 49.33 |
| Community-level media exposure | | |
| Low | 3514 | 46.30 |
| High | 4076 | 53.70 |

Note; * = Catholic, traditional, and other.

## Factors associated with knowledge of ORS packet or pre-packaged liquids for the management of diarrhea in Ethiopia; 2016

**Random effect analysis.** The values of MOR, ICC, and PCV revealed there was a variation of knowledge of ORS packet or pre-packaged liquids between clusters/communities. The ICC in the null model revealed that about 31.2% of the variation in knowledge of ORS packet or pre-packaged liquids was due to differences between communities/clusters. In addition, the MOR value in the null model which was 3.29 revealed that when we randomly select mothers from two clusters, mothers from a high-risk cluster (clusters with no knowledge of ORS packet or pre-packaged liquids) had 3.29 times more likely to lack knowledge about ORS packet or pre-packaged liquids as compared to mothers from a low-risk cluster (clusters with knowledge about ORS packet or pre-packaged liquids). Moreover, the highest PCV value in the final model (model 3) revealed that about 51.4% of the variation in knowledge of ORS packet or pre-packaged liquids was explained by both individual and community-level factors. Deviance was used for model fitness and the model with the lowest deviance (model 3) was selected as the best-fitted model (Table 2). Therefore, model 2, which incorporates both individual and

**Table 2. Random effect analysis and model fitness for the assessment of factors associated with knowledge of ORS packet or pre-packaged liquids.**

| Parameter | Null model | Model 1 | Model 2 | Model 3 |
|---|---|---|---|---|
| Community-level variance (SE) | 1.489 (0.183) | 1.127 (0.147) | 0.731 (0.108) | 0.724 (0.105) |
| ICC (%) | 31.2 | 25.5 | 18.2 | 18 |
| MOR | 3.29[2.79–3.70] | 2.74[2.43–3.15] | 2.25[2.02–2.56] | 2.24[2.01–2.54] |
| PCV (%) | Reference | 24.3 | 50.9 | 51.4 |
| Model fitness | | | | |
| Deviance [-2LL] | 8750.54 | 8495.84 | 8503.62 | 8336.12 |

**Table 3. Multilevel regression analysis of factors associated with knowledge of ORS packet or pre-packaged liquids in Ethiopia; 2016.**

| Variables | Null model [AOR 95% CI] | Model 1 [AOR 95% CI] | Model 2 [AOR 95% CI] | Model 3 [AOR 95% CI] |
|---|---|---|---|---|
| Maternal age (years) | | | | |
| 15–24 | | 1.00 | | 1.00 |
| 25–34 | | 1.34(1.04–1.72) | | 1.29(0.98–1.74) |
| 35–49 | | 1.20(0.91–1.58) | | 1.20(0.91–1.57) |
| Maternal education | | | | |
| No formal education | | 1.00 | | 1.00 |
| Primary education | | 1.36(1.08–1.73) | | 1.32(1.04–1.68) * |
| Secondary education | | 2.51(1.52–4.15) | | 2.14(1.29–3.55) * |
| Tertiary & above education | | 5.59(2.37–13.15) | | 4.43(1.81–10.84) * |
| Religion | | | | |
| Orthodox Christian | | 1.00 | | 1.00 |
| Protestant | | 0.97(0.71–1.31) | | 1.22(0.87–1.73) |
| Muslim | | 1.10(.81–1.49) | | 1.08(0.76–1.52) |
| Others | | 0.57(0.31–1.07) | | 0.74(0.41–1.34) |
| Household wealth status | | | | |
| First | | 1.00 | | 1.00 |
| Second | | 0.83(0.62–1.10) | | 0.93(0.69–1.24) |
| Middle | | 0.93(0.70–1.23) | | 1.05(0.78–1.42) |
| Fourth | | 1.04(0.77–1.41) | | 1.19(0.87–1.63) |
| Higher | | 1.72(1.16–2.54) | | 1.48(0.95–2.31) |
| Employment status | | | | |
| Employed | | 1.25(0.99–1.57) | | 1.24(0.99–1.56) |
| Not employed | | 1.00 | | 1.00 |
| Distance from the health facility | | | | |
| Big problem | | 1.00 | | 1.00 |
| Not big problem | | 1.04(0.81–1.33) | | 0.99(0.77–1.27) |
| Number of under five children | | | | |
| 0 | | 1.00 | | 1.00 |
| 1–2 | | 1.27(0.79–2.05) | | 1.28(0.79–2.07) |
| 3–6 | | 1.15(0.69–1.91) | | 1.11(0.67–1.85) |
| Media exposure | | | | |
| No | | 1.00 | | 1.00 |
| Yes | | 1.57(1.25–1.96) | | 1.57(1.26–1.97) * |
| Residence | | | | |
| Urban | | | 1.00 | 1.00 |
| Rural | | | 0.27(0.17–0.44) | 0.53(0.30–0.95) * |
| Region | | | | |
| Tigray | | | 1.00 | 1.00 |
| Afar | | | 0.83(0.49–1.40) | 0.83(0.44–1.58) |
| Amhara | | | 0.19(0.12–0.31) | 0.20(0.12–0.31) * |
| Oromia | | | 0.15(0.10–0.23) | 0.14(0.08–0.22) * |
| Somalia | | | 0.87(0.48–1.59) | 0.90(0.45–1.80) |
| Benishangul | | | 0.68(0.41–1.11) | 0.61(0.36–1.05) |
| SNNPR | | | 0.19(0.12–0.29) | 0.16(0.10–0.26) * |
| Gambela | | | 0.26(0.16–0.42) | 0.21(0.12–0.37) * |
| Harari | | | 1.28(0.75–2.16) | 1.21(0.65–2.25) |
| Addis Ababa | | | 0.49(0.23–1.04) | 0.35(0.16–0.76) * |
| Dire Dawa | | | 0.44(0.21–0.93) | 0.41(0.19–0.92) * |

(*Continued*)

**Table 3.** (Continued)

| Variables | Null model [AOR 95% CI] | Model 1 [AOR 95% CI] | Model 2 [AOR 95% CI] | Model 3 [AOR 95% CI] |
|---|---|---|---|---|
| Community poverty level | | | | |
| Low | | | 1.00 | 1.00 |
| High | | | 0.80(0.61–1.06) | 0.90(0.66–1.23) |
| Community illiteracy level | | | | |
| Low | | | 1.00 | 1.00 |
| High | | | 0.57(0.44–0.74) | 0.66(0.51–0.87) * |
| Community-level media exposure | | | | |
| Low | | | 1.00 | 1.00 |
| High | | | 0.99(0.77–1.27) | 0.81(0.63–1.06) |

Note;

* = P<0.05,

AOR = Adjusted Odds Ratio, CI = Confidence Interval.

community level factors, was used for the assessment of factors that were associated with knowledge of ORS packet or pre-packaged liquids.

**Fixed effects analysis.** Variables with a p-value <0.20, in the bivariable analysis, were eligible for the multivariable analysis. All variables except sex of household head and marital status were incorporated in the multivariable analysis. Both individual-level variables (maternal education and media exposure) and community-level variables (residence, community illiteracy level, and region) were significantly associated with knowledge of ORS packet or pre-packaged liquids (p<0.05). Mothers who had primary education (AOR = 1.34; 95%CI: 1.06–1.68), secondary education (AOR = 2.14; 95%CI: 1.29–3.55), and tertiary and above education (AOR = 4.48; 95%CI: 1.1.81–10.84) had higher odds of knowledge of ORS packet or pre-packaged liquids as compared to those who did not attend formal education. The odds of knowledge of ORS packet or pre-packaged liquids were 1.57 (AOR = 1.57; 95%CI: 1.26–1.97) times higher among mothers who were exposed to media as compared to their counterparts. Mothers from the rural area had 47% (AOR = 0.53; 95%CI: 0.30–0.95) lower odds of knowledge of ORS packet or pre-packaged liquids as compared to their counterparts. Regarding community illiteracy level, mothers from communities with higher illiteracy level had 34% (AOR = 0.66; 95%CI: 0.51–0.87) lower odds of knowledge of ORS packet or pre-packaged liquids as compared to their counterparts. In addition, region was another community-level factor that was associated with the odds of knowledge of ORS packet or pre-packaged liquids (Table 3).

## Spatial analysis of knowledge of ORS packet or pre-packaged liquids in Ethiopia

**Spatial autocorrelation.** Knowledge of ORS packet or pre-packaged liquids for the management of diarrhea was not random across Ethiopia (with Moran's Index = 0.46 and p-value <0.001) (Fig 1).

**Spatial interpolation.** Spatial interpolation was done using the Kriging interpolation method. As shown in Fig 2, most parts of the Amhara, SNNPR, and Oromia regions had the highest predicted proportion of mothers with no knowledge about ORS packet or pre-packaged liquids for the management of diarrhea. While Tigray, Somali, Benishangul, Addis Ababa, Dire Dawa, and Harari had the lowest predicted proportions of mothers with no knowledge about ORS packet or pre-packaged liquids for the management of diarrhea (Fig 2).

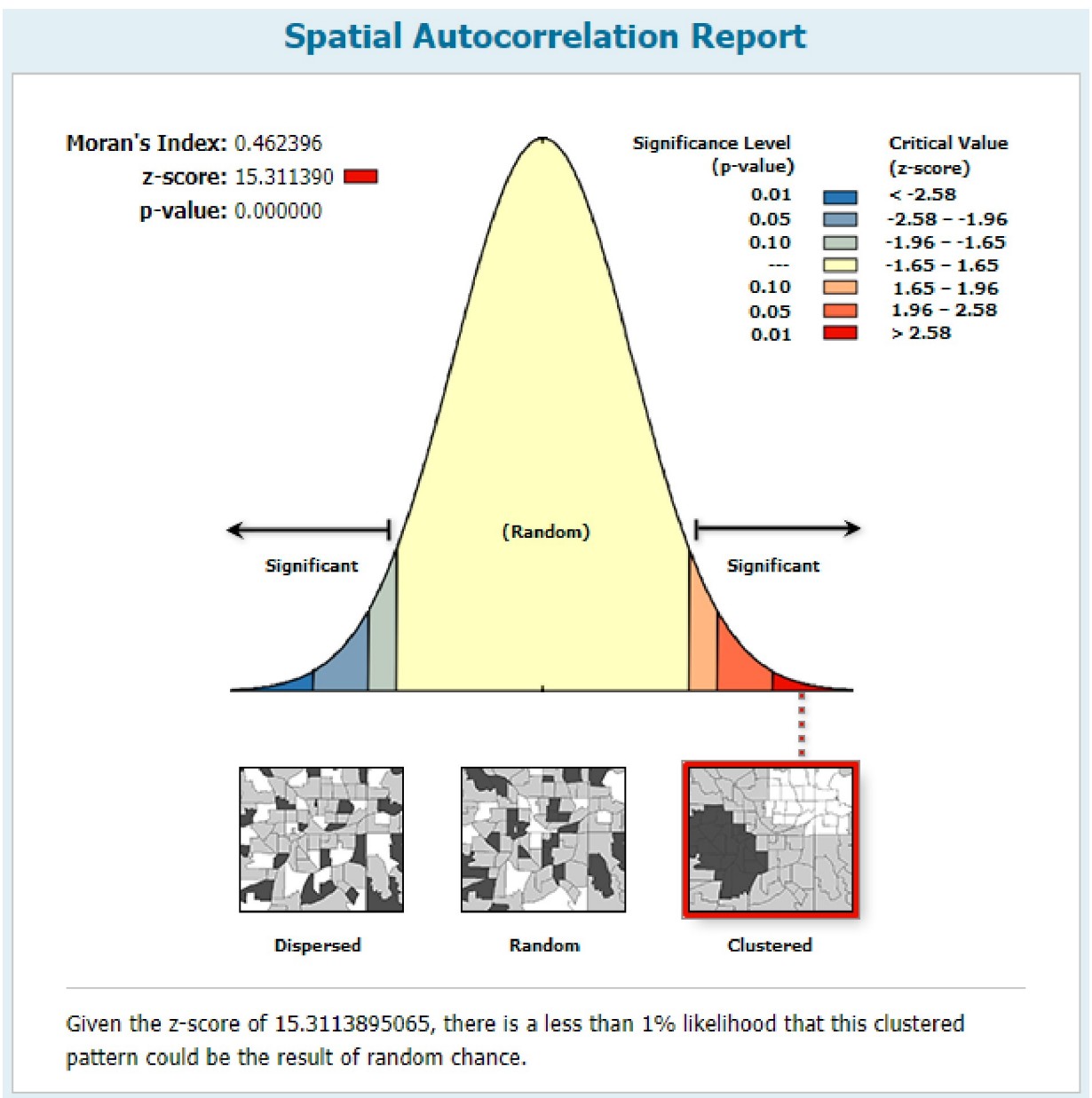

**Fig 1. Spatial autocorrelation result showing clustering of knowledge of ORS packet or pre-packaged liquids for the management of diarrhea in Ethiopia; map produced using Arc GIS version 10.3.**

**Hotspot and cold spot analysis.**    Fig 3 revealed Hot spot and cold spot analysis using Getis-Ord Gi* statistics. Regions with red-colored clustered points (Amhara, Oromia, most parts of SNNPR, some parts of southwestern Afar) had significantly higher rates of lack of knowledge regarding ORS packet or pre-packaged liquids for the management of diarrhea. However, the blue-colored clustered points revealed areas with significantly lower rates of lack of knowledge about ORS packet or pre-packaged liquids. These were found in Tigray, Addis Ababa, Harari, Dire Dawa, Somalia, and western parts of Benishangul (Fig 3).

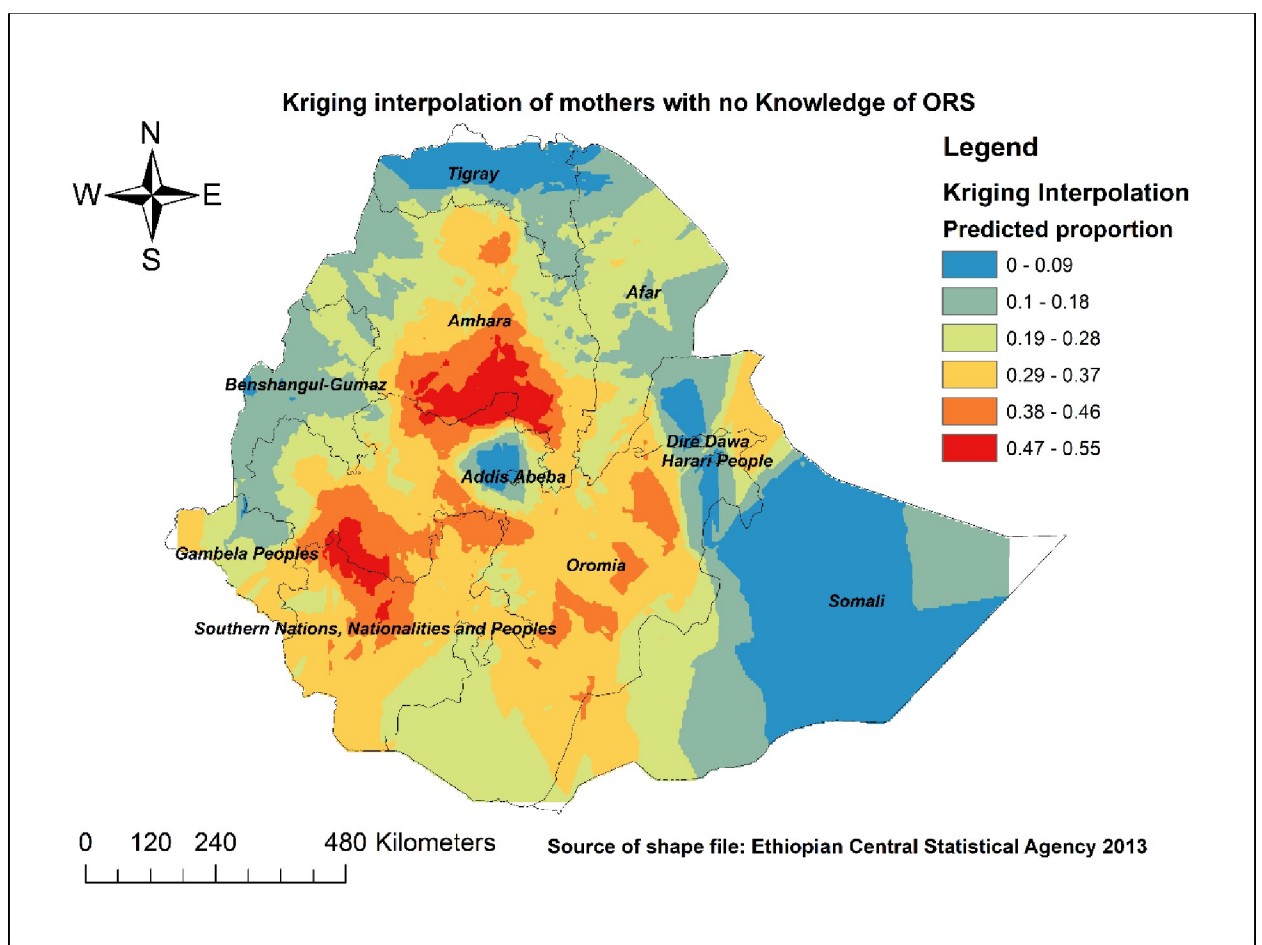

**Fig 2. Kriging interpolation result of knowledge of ORS packet or pre-packaged liquids for the management of diarrhea in Ethiopia; map produced using Arc GIS version 10.3.**

**SaTScan analysis using the Bernoulli based model.** We performed a SaTScan analysis to identify the primary and secondary clusters. The SaTScan analysis identified 192 significant clusters. Of these, 127 clusters were primary or the most likely clusters and the rest were secondary clusters. As shown in Fig 4, the primary clusters spatial window was located in SNNPR, most parts of the Oromia region and eastern parts of Gambela centered at 5.918058 N, 37.291418 E of location with 363.45 km radius, LLR of 91.47, and p value<0.001. The RR was 1.86 and this means that, relative to mothers from outside the spatial window, mothers inside the spatial window had a 1.86 times greater chance of lack of knowledge about ORS packets or pre-packaged liquid. The secondary clusters spatial window was located near areas of the border between Amhara, Oromia, and Afar regions, as well as at the border of Amhara and Tigray regions (Fig 4).

## Discussion

This study aimed to assess the spatial variations and associated factors of knowledge of ORS packet or pre-packaged liquids for the management of diarrhea in Ethiopia. In the multilevel analysis, maternal education and media exposure were among the individual-level factors that were associated with knowledge of ORS packet or pre-packaged liquids. Among the

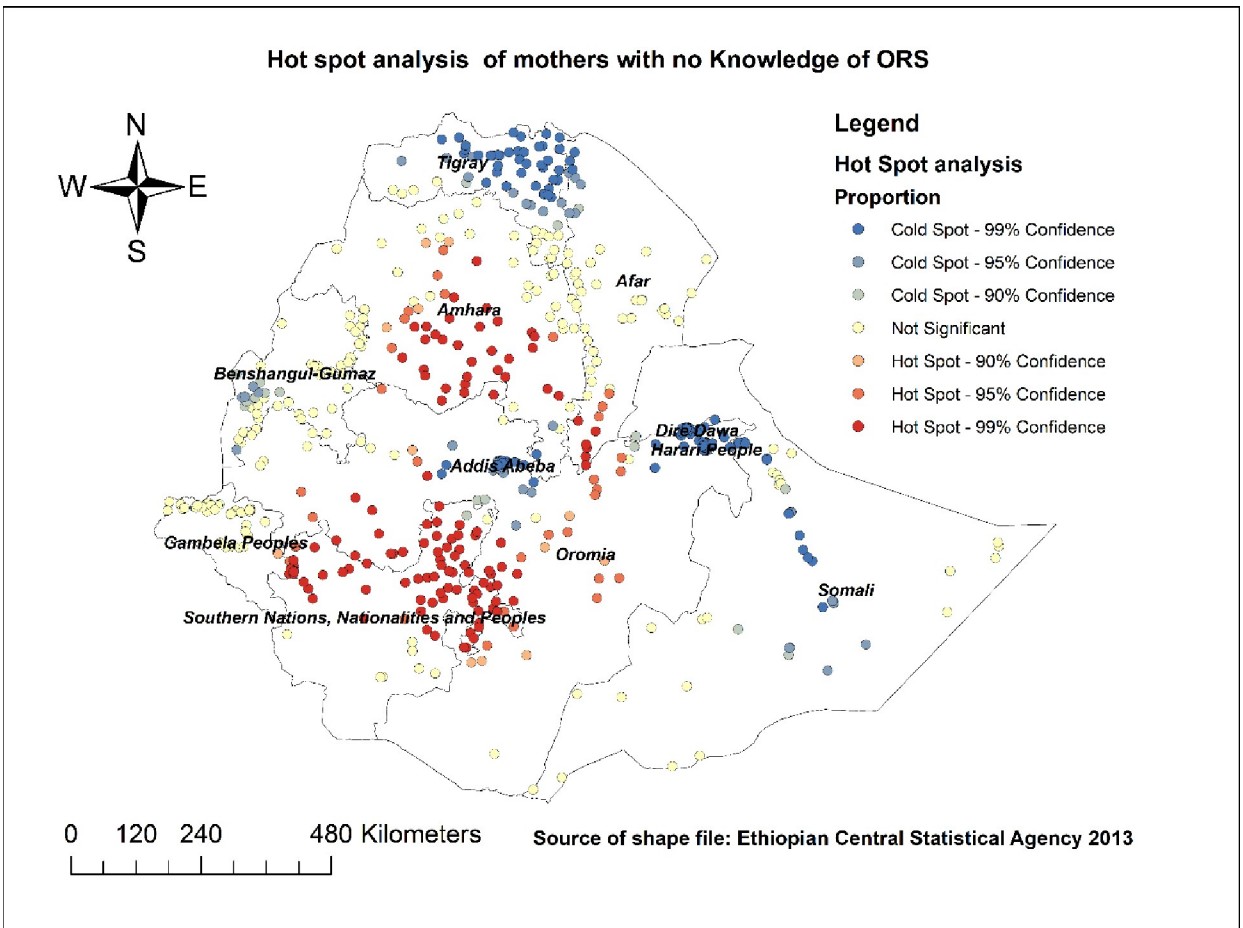

**Fig 3. Hot spot and cold spot analysis of knowledge regarding ORS packet or pre-packaged liquids for the management of diarrhea in Ethiopia; map produced using Arc GIS version 10.3.**

community-level factors, residence, community illiteracy level, and region were significantly associated with knowledge of ORS packet or pre-packaged liquids.

Mothers having primary, secondary, and tertiary and higher education had higher odds of knowledge of ORS packet or pre-packaged liquids as compared to those who had not attended formal education. Besides, mothers who were from communities with higher women's illiteracy level had lower odds of knowledge of ORS packet or pre-packaged liquids. This finding is supported by a study done in India [11], Iran [15], Nigeria [14], and Ethiopia [13]. This might be since as the educational level of women increases the level of awareness and knowledge on ORS packet or pre-packaged liquids increases. Besides, women from communities with higher education might be more likely to have a greater understanding of effective diarrhea management mechanisms, including the ORS packet or pre-packaged liquids.

In this study, media exposure was associated with knowledge of ORS packet or pre-packaged liquids. Women who had been exposed to media had higher odds of knowledge of ORS packet or pre-packaged liquids as compared to their counterparts. This is in concordance with a study in India [12]. It is obvious that mothers with media exposure might be exposed to management options of acute diarrhea in under-five children and had knowledge regarding ORS packet or pre-packaged liquids.

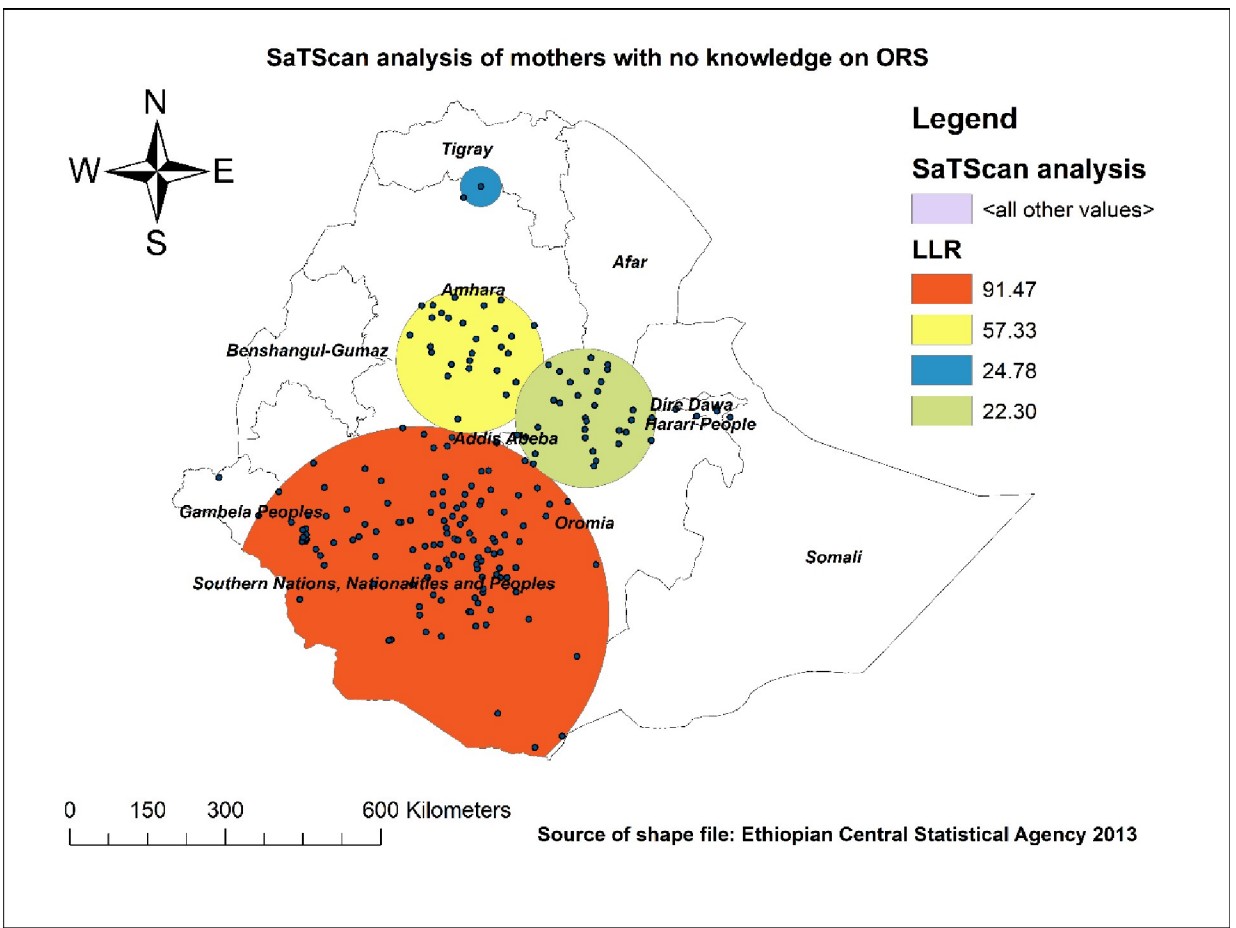

**Fig 4. SaTScan analysis of knowledge regarding ORS packet or pre-packaged liquids for the management of diarrhea in Ethiopia; map produced using Arc GIS version 10.3.**

Moreover, the study at hand also revealed that being living in a rural area was associated with lower odds of knowledge of ORS packet or pre-packaged liquids as compared to their counterparts. This is consistent with a study done in Ethiopia [16]. This is due to urban women have access to a wider variety of services and sources of information, including ORS packets or pre-packaged liquids, than do rural women in Ethiopia.

The study found that the spatial distribution of knowledge of ORS packet or pre-packaged liquids was not random in Ethiopia. The primary clusters spatial window was located in SNNPR, most parts of the Oromia region, and eastern parts of Gambela. The secondary clusters spatial window was found in near areas of the border between Amhara, Oromia, and Afar regions, as well as at the border of Amhara and Tigray regions. This finding is supported by the multilevel analysis conducted in this study. This might be due to the sociocultural difference between women in these regions. Moreover, this might be because of the difference in the level of awareness and level of education between women in these regions.

This study had strengths as well as limitations. Since it is based on nationally representative data, it is appropriate for giving direction for policymakers and program planners to plan intervention strategies. Besides, for a better estimate of parameters, we used a multilevel

analysis. Moreover, the detection of hot spot areas using spatial analysis may help policy-makers to decide and plan intervention mechanisms accordingly, by giving priority to the identified hot spot areas. However, it is difficult to investigate causality among dependent and independent variables due to the cross-sectional nature of the data.

## Conclusion

In this study knowledge on ORS packet or pre-packaged liquids was not random across Ethiopia. The primary clusters spatial window was located in SNNPR, most parts of Oromia region, and eastern parts of Gambela, and the secondary clusters spatial window was found in near areas of the border between Amhara, Oromia, and Afar region, as well as at the border of Amhara and Tigray region. In the multilevel analysis, maternal education, media exposure, residence, community illiteracy level, and region were significantly associated with knowledge of ORS packet or pre-packaged liquids. Therefore, it is better to give special emphasis to women who had no formal education and who are from communities with no education, as well as those who came from rural areas. Also, those hot spot regions should be given attention. Moreover, distributing information regarding this public health important issue through different media is necessary.

## Acknowledgments

Our deepest gratitude and appreciation go to the measure DHS program for allowing us to use the data set.

## Author Contributions

**Conceptualization:** Achamyeleh Birhanu Teshale.

**Data curation:** Achamyeleh Birhanu Teshale, Getayeneh Antehunegn Tesema, Zemenu Tadesse Tessema.

**Formal analysis:** Achamyeleh Birhanu Teshale, Getayeneh Antehunegn Tesema, Zemenu Tadesse Tessema.

**Investigation:** Achamyeleh Birhanu Teshale, Getayeneh Antehunegn Tesema, Zemenu Tadesse Tessema.

**Methodology:** Achamyeleh Birhanu Teshale, Getayeneh Antehunegn Tesema, Zemenu Tadesse Tessema.

**Resources:** Achamyeleh Birhanu Teshale, Getayeneh Antehunegn Tesema, Zemenu Tadesse Tessema.

**Software:** Achamyeleh Birhanu Teshale, Getayeneh Antehunegn Tesema, Zemenu Tadesse Tessema.

**Validation:** Achamyeleh Birhanu Teshale, Getayeneh Antehunegn Tesema, Zemenu Tadesse Tessema.

**Visualization:** Achamyeleh Birhanu Teshale, Getayeneh Antehunegn Tesema, Zemenu Tadesse Tessema.

**Writing – original draft:** Achamyeleh Birhanu Teshale, Getayeneh Antehunegn Tesema, Zemenu Tadesse Tessema.

**Writing – review & editing:** Achamyeleh Birhanu Teshale, Getayeneh Antehunegn Tesema, Zemenu Tadesse Tessema.

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
