## [Decision Letter · Decision Letter 0]

26 Jan 2021

PONE-D-20-36968

Spatial variations and associated factors of Knowledge of ORS packet or pre-packaged liquids for the management of diarrhea among women of reproductive age in Ethiopia: a spatial and multilevel analysis

PLOS ONE

Dear Dr. Teshale,

Thank you for submitting your manuscript to PLOS ONE. After careful consideration, we feel that it has merit but does not fully meet PLOS ONE’s publication criteria as it currently stands. Therefore, we invite you to submit a revised version of the manuscript that addresses the points raised during the review process.

We look forward to receiving your revised manuscript.

Kind regards,

Vijayaprasad Gopichandran

Academic Editor

PLOS ONE

Journal Requirements:

2. We note that Figures 1-4 in your submission contain map images which may be copyrighted. All PLOS content is published under the Creative Commons Attribution License (CC BY 4.0), which means that the manuscript, images, and Supporting Information files will be freely available online, and any third party is permitted to access, download, copy, distribute, and use these materials in any way, even commercially, with proper attribution. For these reasons, we cannot publish previously copyrighted maps or satellite images created using proprietary data, such as Google software (Google Maps, Street View, and Earth). For more information, see our copyright guidelines: http://journals.plos.org/plosone/s/licenses-and-copyright.

2.1.    You may seek permission from the original copyright holder of Figure 1-4 to publish the content specifically under the CC BY 4.0 license. 

2.2.    If you are unable to obtain permission from the original copyright holder to publish these figures under the CC BY 4.0 license or if the copyright holder’s requirements are incompatible with the CC BY 4.0 license, please either i) remove the figure or ii) supply a replacement figure that complies with the CC BY 4.0 license. Please check copyright information on all replacement figures and update the figure caption with source information. If applicable, please specify in the figure caption text when a figure is similar but not identical to the original image and is therefore for illustrative purposes only.

- https://bmcpublichealth.biomedcentral.com/articles/10.1186/s12889-020-09541-4

- https://www.researchsquare.com/article/rs-15540/v1

In your revision ensure you cite all your sources (including your own works), and quote or rephrase any duplicated text outside the methods section. Further consideration is dependent on these concerns being addressed.

Reviewers' comments:

Reviewer's Responses to Questions

**Comments to the Author**

1. Is the manuscript technically sound, and do the data support the conclusions?

Reviewer #1: Partly

2. Has the statistical analysis been performed appropriately and rigorously? 

Reviewer #1: I Don't Know

3. Have the authors made all data underlying the findings in their manuscript fully available?

Reviewer #1: No

4. Is the manuscript presented in an intelligible fashion and written in standard English?

Reviewer #1: No

5. Review Comments to the Author

Reviewer #1: The article describes the spatial variations in the knowledge of ORS packet (therapy) or Pre-packed liquids among the women of reproductive age group in Ethiopia. I appreciate the team efforts for conducting the robust geo-spatial analysis which potentially aids targeted public health interventions.

Major Issues:

1) The article had a lot of grammatical errors and will need major editing and revision. I would suggest using a copyeditor to improve the flow and readability once the revision is ready for submission.

Minor issues:

2) L 56- I would suggest to reconsider the using brand name “LEM LEM”. Using brand names in the scientific communication is not advisable in general.

3) L 58-61 Two consecutive sentences start with “Different literature”, consider rephrasing it

EDHS is a methods and reports are available in the public domain. The section titled “Sampling technique, sample size, and population” is not clear. Please rewrite this section to give a better experience for readers.

I suggest to give references for data source and methodology.

4) L 94-97 The definition of “Knowledge of ORS” which was the outcome of the study is unclear. I suggest to mention the exact definition used in the Ethiopia DHS, 2016.

5) I suggest to present the multilevel analysis first followed spatial analysis under methods as well as in result section. Describe the spatial analysis in detail (example: assumptions used, methods used, variables, defining threshold level, clustering etc.). The detailed description of analysis would help others to reproduce the analysis.

6) Under the discussion section, discuss how this analysis would be useful in implementing public health interventions.

6. PLOS authors have the option to publish the peer review history of their article (what does this mean?). If published, this will include your full peer review and any attached files.

Reviewer #1: No

---

## [Author Response · Author response to Decision Letter 0]

12 Feb 2021

Date: February 12, 2021

Point by point response 

Title: Spatial variations and associated factors of Knowledge of ORS packet or pre-packaged liquids for the management of diarrhea among women of reproductive age in Ethiopia: a spatial and multilevel analysis

Manuscript number: PONE-D-20-36968

Dear editor and reviewer, thank you for the valuable comments you raised for the betterment of our manuscript. Really, the comments were important and we put the point-by-point response below. We also incorporated the comments and suggestions in the revised manuscript. 

Thank you in advance!

Response to editor comment 

Author’s response: Thank you. We ensure that the revised manuscript meets the PLOS ONE's style requirements, including those for file naming. 

2. We note that Figures 1-4 in your submission contain map images which may be copyrighted. All PLOS content is published under the Creative Commons Attribution License (CC BY 4.0), which means that the manuscript, images, and Supporting Information files will be freely available online, and any third party is permitted to access, download, copy, distribute, and use these materials in any way, even commercially, with proper attribution. For these reasons, we cannot publish previously copyrighted maps or satellite images created using proprietary data, such as Google software (Google Maps, Street View, and Earth). 

We require you to either (1) present written permission from the copyright holder to publish these figures specifically under the CC BY 4.0 license, or (2) remove the figures from your submission.

Author’s response: Thank you for the comment. These figures are not copyrighted from other sources rather they are our findings using Arc-GIS version 10.3 and SaTScan version 9.6 statistical softwares. After getting the shape file of Ethiopia in the website https://africaopendata.org/dataset/ethiopia-shapefiles, we generate the figures using the GPs (latitude and longitude) data and the outcome variable using ArcGIS version 10.3 and SaTScan version 9.6 statistical softwares. So all the figures are not copyrighted from other source rather we generate using the software.

- https://bmcpublichealth.biomedcentral.com/articles/10.1186/s12889-020-09541-4

- https://www.researchsquare.com/article/rs-15540/v1

In your revision ensure you cite all your sources (including your own works), and quote or rephrase any duplicated text outside the methods section. Further consideration is dependent on these concerns being addressed.

Author’s response: Thank you for the comment. We have considered your comment in the revised manuscript; we avoid the overlapped texts with the given published works.

Response to reviewer comment (Reviewer #1)

Major Issues:

1. The article had a lot of grammatical errors and will need major editing and revision. 

Author’s response: We extensively edited the manuscript for grammatical errors, by consulting our colleagues and language experts who had many years’ experience in the area of literature at University of Gondar. A copy of our manuscript showing the changes is indicated by using track changes (See the track-changed manuscript). 

Minor issues:

2. L 56- I would suggest to reconsider the using brand name “LEM LEM”. Using brand names in the scientific communication is not advisable in general.

Author’s response: Thank you. We have consider your issue in the revised manuscript.

3. L 58-61 Two consecutive sentences start with “Different literature”, consider rephrasing it

Author’s response: Thank you. We rephrase these sentences in the revised manuscript. 

EDHS is a methods and reports are available in the public domain. The section titled “Sampling technique, sample size, and population” is not clear. Please rewrite this section to give a better experience for readers. I suggest to give references for data source and methodology.

Author’s response: Really, thank you for the important concern you raised. We have rewritten the section “Sampling technique, sample size, and population”; we have putted these in the data source section of the method and we have incorporated the reference in the revised manuscript. 

4. L 94-97 The definition of “Knowledge of ORS” which was the outcome of the study is unclear. I suggest to mention the exact definition used in the Ethiopia DHS, 2016.

Author’s response: Thank you. We used the DHS guide and the EDHS 2016 report to assess Knowledge about ORS packet or prepackaged liquids. Women had knowledge about ORS packet or pre-packaged liquids if she heard about it or she used it for the management of diarrhea. We have extracted the outcome variable using v208 > 0 & v416 in 1, 2. Therefore, that is why we used that definition (the definition is in line with the EDHS 2016). 

5. I suggest to present the multilevel analysis first followed spatial analysis under methods as well as in result section. Describe the spatial analysis in detail (example: assumptions used, methods used, variables, defining threshold level, clustering etc.).The detailed description of analysis would help others to reproduce the analysis.

Author’s response: Thank you. We consider your comment and put the multilevel analysis first in the method, result, and discussion sections. In addition, we have described the spatial analysis accordingly and we put references for anyone who is interested to know about the assumptions and other information regarding the spatial analysis. 

6. Under the discussion section, discuss how this analysis would be useful in implementing public health interventions.

Author’s response: Thank you. We have discussed the public health implication of our study in the revised manuscript (see the last paragraph of the discussion section).

---

## [Editor Report · Decision Letter 1]

15 Feb 2021

Spatial variations and associated factors of Knowledge of ORS packet or pre-packaged liquids for the management of diarrhea among women of reproductive age in Ethiopia: a spatial and multilevel analysis

PONE-D-20-36968R1

Dear Dr. Teshale,

We’re pleased to inform you that your manuscript has been judged scientifically suitable for publication and will be formally accepted for publication once it meets all outstanding technical requirements.

Kind regards,

Vijayaprasad Gopichandran

Academic Editor

PLOS ONE
---

## [Editor Report · Acceptance letter]

18 Mar 2021

PONE-D-20-36968R1 

Spatial variations and associated factors of Knowledge of ORS packet or pre-packaged liquids for the management of diarrhea among women of reproductive age in Ethiopia: a spatial and multilevel analysis 

Dear Dr. Teshale:

I'm pleased to inform you that your manuscript has been deemed suitable for publication in PLOS ONE. Congratulations! Your manuscript is now with our production department. 

Kind regards, 

on behalf of

Dr. Vijayaprasad Gopichandran 

Academic Editor

PLOS ONE